# The Impact of Fast Radiation on the Phylogeny of *Bactrocera* Fruit Flies as Revealed by Multiple Evolutionary Models and Mutation Rate-Calibrated Clock

**DOI:** 10.3390/insects13070603

**Published:** 2022-06-30

**Authors:** Federica Valerio, Nicola Zadra, Omar Rota-Stabelli, Lino Ometto

**Affiliations:** 1Department of Biology and Biotechnology, University of Pavia, 27100 Pavia, Italy; federica.valerio02@universitadipavia.it; 2Research and Innovation Centre, Fondazione Edmund Mach, 38010 San Michele all’Adige, Italy; zadranic@gmail.com (N.Z.); omar.rotastabelli@unitn.it (O.R.-S.); 3Department of Cellular, Computational and Integrative Biology—CIBIO, University of Trento, 38123 Trento, Italy; 4Center Agriculture Food Environment—C3A, University of Trento, 38010 San Michele all’Adige, Italy

**Keywords:** *Bactrocera*, phylogenomics, phylogeny, dating, incomplete lineage sorting, *Bactrocera dorsalis*, *Bactrocera tryoni*

## Abstract

**Simple Summary:**

Comparative approaches are widely used to investigate the traits that underlie particular biological and ecological traits. They are effective, however, only if we know the correct relationships among the species under consideration. Here, we aimed at reconstructing a robust phylogeny for selected true fruit flies of the genus *Bactrocera*, which are well known as important agricultural pests worldwide. Existing phylogenetic inferences are still ambiguous, especially concerning the relationship between the two major pests *B. dorsalis* and *B. tryoni*. In this study, we employed a genome-scaled dataset and different models of molecular evolution to reconstruct the phylogenetic relationships of ten *Bactrocera* species and two outgroups and further date their divergence times. The resulting phylogeny fully supports *B. dorsalis* as more closely related to *B. latifrons* than to *B. tryoni*, opposite to what was supported by previous works. This incongruence likely derives from a fast divergence of these lineages, as revealed by our clock analysis, which can lead to conflicting results when using few genetic markers. Our results thus highlight the utility of using large datasets and of exploring different evolutionary models to study the evolutionary history of species of economic importance.

**Abstract:**

Several true fruit flies (Tephritidae) cause major damage to agriculture worldwide. Among them, species of the genus *Bactrocera* are extensively studied to understand the traits associated with their invasiveness and ecology. Comparative approaches based on a reliable phylogenetic framework are particularly effective, but several nodes of the *Bactrocera* phylogeny are still controversial, especially concerning the reciprocal affinities of the two major pests *B. dorsalis* and *B. tryoni*. Here, we analyzed a newly assembled genomic-scaled dataset using different models of evolution to infer a phylogenomic backbone of ten representative *Bactrocera* species and two outgroups. We further provide the first genome-scaled inference of their divergence by calibrating the clock using fossil records and the spontaneous mutation rate. The results reveal a closer relationship of *B. dorsalis* with *B. latifrons* than to *B. tryoni*, contrary to what was previously supported by mitochondrial-based phylogenies. By employing coalescent-aware and heterogeneous evolutionary models, we show that this incongruence likely derives from a hitherto undetected systematic error, exacerbated by incomplete lineage sorting and possibly hybridization. This agrees with our clock analysis, which supports a rapid and recent radiation of the clade to which *B. dorsalis*, *B. latifrons* and *B. tryoni* belong. These results provide a new picture of *Bactrocera* phylogeny that can serve as the basis for future comparative analyses.

## 1. Introduction

Tephritidae, commonly known as true fruit flies, are an incredibly diverse group of phytophagous insects. This family includes more than 5000 species in over 500 genera [1], making it one of the largest among Diptera. Several of these species cause extensive damage to agriculture worldwide, since females lay eggs within the fruit flesh (i.e., pericarp) and the larvae rapidly develop in the fruit, eating it and inducing bacterial and fungal decay. The accompanying economic impacts are considerable: for instance, the olive fruit fly, *Bactrocera oleae* (Rossi), has been estimated to cause an annual economic loss of approximately $800 million in the Mediterranean basin alone [2]. Moreover, such impact is exacerbated by these species’ ability to invade new areas and establish viable populations. For example, *Bactrocera dorsalis* and *Bactrocera latifrons*, both native to South-East Asia, have recently expanded their distribution range in the Hawaiian Islands [3,4], in several sub-Saharan African countries [5,6,7,8,9,10,11,12], and also in Europe [9,13,14], where they are posing an immediate or potential threat to local agriculture.

Most relevant tephritid pests belong to five genera: *Anastrepha* Schiner, *Ceratitis* MacLeay, *Rhagoletis* Loew, *Zeugodacus* Hendel and *Bactrocera* Macquart. The genus *Bactrocera* is the most economically important and comprises over 450 species [15], at least 50 of which are considered important pests [16,17]. Identifying the key ecological and biological traits relevant for their invasiveness and host preference is extremely important to drive effective control strategies. To this aim, comparative approaches based on a reliable phylogenetic framework are particularly effective, as they allow to trace the evolution of species-specific and shared traits and evaluate whether and to what extent control measures may be applied in related species [18]. The *Bactrocera* genus is subdivided into subgenera, species groups and species complexes, and the relationships between these groups have been repeatedly revised (see, e.g., [19,20,21,22,23,24,25]). Different molecular phylogenies of *Bactrocera* resulted in different relationships depending on the type (nuclear and/or mitochondrial) and number of markers, and on the number of taxa analyzed [23,24,25,26,27,28,29]. For instance, based on mtDNA data, *B. dorsalis* is usually considered to be more closely related to *B. tryoni* than to *B. latifrons* (e.g., [30,31]), but recent studies suggested a poorly supported closer relationship between *B. dorsalis* and *B. latifrons* [24,26,32] (Figure 1a). This uncertainty may also be important for studies relevant for their management, since different phylogenetic reconstructions may affect the interpretation of comparative morphological and genomic studies that focus on pest-related biological traits.

Here, we employed a rigorous phylogenomic approach to infer the phylogeny of ten representative *Bactrocera* species, including seven of the nine major agricultural pests of worldwide importance (see Table 1 in [16]), and we calibrated a relaxed clock to estimate their divergences. Rather than resolving the full *Bactrocera* phylogeny, our aim is to provide a robust phylogenetic and chronological backbone that can clarify the yet unresolved (*B. dorsalis*, *B. tryoni*, *B. latifrons*) clade, and which can then serve as basis for powerful comparative analyses. Our dataset also includes species characterized by distinct diets, which range from polyphagy (e.g., *B. dorsalis*) to monophagy (e.g., *B. oleae*), and, therefore, understanding the evolutionary history of this group may also provide precious information on the mode and tempo of the evolution of their biology and ecology.

## 2. Materials and Methods

### 2.1. Datasets

In our analysis, we included *Bactrocera* and Dacini species for which genomic and/or transcriptomic resources were available: *Bactrocera jarvisi* (Tryon), *Bactrocera oleae* (Rossi), *Bactrocera minax* (Enderlein), *Bactrocera bryoniae* (Tryon), *Bactrocera correcta* (Bezzi), *Bactrocera dorsalis* (Hendel), *Bactrocera latifrons* (Hendel), *Bactrocera musae* (Tryon), *Bactrocera tryoni* (Froggatt), *Bactrocera zonata* (Saunders) and *Zeugodacus cucurbitae* (Coquillett). In particular, we analyzed datasets of coding sequences (CDS) and of RNA (transcript) sequences for *Ceratitis capitata* [33], *Z. cucurbitae* [34], *B. dorsalis* [35], *B. latifrons* (NCBI accession: MIMC00000000.1), *B. minax* [36] and *B. oleae* [37]. For the six remaining *Bactrocera* species (*B. bryoniae*, *B. correcta*, *B. jarvisi*, *B. musae*, *B. tryoni* and *B. zonata*), we downloaded the available RNA-Seq raw reads (see Appendix A for SRA accession numbers) and assembled the corresponding transcriptomes using default parameters with Trinity v. 2.7.0 [38].

### 2.2. Orthologous Gene Set Identification and Alignment

Orthologs across the ten *Bactrocera* species and the outgroups *Z. cucurbitae* and *C*. *capitata* were identified using a reciprocal-best-hit approach using pairwise BLASTn searches [39] between the *C. capitata* CDS sequences and each of the other datasets. Putative 1:1 orthologs were first aligned using MAFFT [40] and any incomplete codon (based on the *C. capitata* sequence) was removed. We then re-aligned the ortholog sets using the PRANK algorithm [41] implemented in the tool TranslatorX [42]. We minimized bias in our datasets by 1) removing alignments containing sequences with internal stop codons and 2) using a custom perl script to remove problematic and ambiguous alignment regions [43]. Using this pipeline, we ultimately identified 110 orthologous gene sets across all twelve species (single alignments are available as Appendix A).

We concatenated orthologs to generate an alignment of 189,891 nucleotides (nts) and then translated it using the standard genetic code to obtain an alignment of 63,297 amino acids (aas). We further generated an alignment of 24,885 nts containing only 4-fold degenerate sites retrieved using MEGA7 [44].

### 2.3. Phylogenetic Analyses

We inferred phylogenetic relationships using both a maximum likelihood (ML) and a Bayesian approach. To explore possible sources of systematic errors, we employed homogenous, heterogeneous and coalescent-aware models of evolution.

We ran ML analyses on both the concatenated aa alignment using the PROTGAMMAGTR model, and on the nt alignment partitioned in first, second and third (1 + 2 + 3) position using the GAMMAGTR model implemented in RAxML [45]. We also ran an ML analysis based on the 4-fold degenerate site alignment using the GAMMAGTR model. In all cases, node support was calculated by the rapid bootstrap feature of RAxML (100 replicates). We also estimated bootstrap supports using the coalescent-aware analysis of ASTRAL [46], which was based on all single ML gene trees obtained by RAxML using the same models of the concatenated analyses for either the nt or the aa sequences. Bootstrap values were estimated by performing either 100 multi-locus bootstrap replicates or gene + site resampling (using the -g option).

The same three datasets, aas, nts (1 + 2 + 3), and 4-fold degenerate sites, were used to run Bayesian analyses in BEAST v. 2.5.1 [47]. The aminoacidic dataset was analyzed with a LG + G4 substitution model. The 4-fold degenerate site dataset and the codon dataset were analyzed with a GTR + G4 substitution model. The codon dataset was split into three partitions, corresponding to the codon positions, setting linked trees across them. We performed additional Bayesian analysis on the aminoacidic dataset using a CAT + GTR model and on the 4-fold degenerate site dataset using the among-site heterogeneous CAT model with gamma distribution with PhyloBayes [48]. In both cases, we ran two independent MCMC chains and checked for convergence using the associated tracecomp and bpcomp commands. For the aminoacidic dataset we let both chains run until parameters were stabilized with *maxdiff* = 0, *reldiff* < 0.2 (except for *loglik*, *alpha*, *stat* and *rrent*, with *reldiff* between 0.62 and 0.77) and the effective sample size was *effsize* > 170 (except for *loglik*, *alpha*, *stat*, *rrent* and *allocent*, with *effsize* between 16 and 119). For the 4-fold degenerate site dataset, both chains ran until parameters were stabilized with *maxdiff* = 0, *reldiff* < 0.25 (except for *stat*, with *reldiff* < 0.4) and the effective sample size was *effsize* > 170 (except for *nmode*, *statalpha*, *kappa* and *allocent*, with *effsize* between 17 and 85). The summary statistics reached stability at the end of the runs and were periodically visualized with the script graphphylo (https://github.com/wrf/graphphylo accessed on 15 April 2022).

Bayesian analyses were also ran using StarBEAST2 [49], which employs a multispecies coalescent method to estimate species trees from multiple sequence alignments, as implemented in the BEAST2 package and according to the tutorial provided by Taming the BEAST2 [50]. For this analysis we used either the nucleotide or the aminoacidic alignments, linking the site models across the gene sets.

BEAST2 and StarBEAST2 analyses had chains run for 10^8^ generations (or until convergence for at least 4 × 10^7^ generations), sampling trees and parameters every 1000 generations and inspecting convergence and likelihood plateauing in Tracer. Posterior consensus trees were generated after discarding the first 10% of generations as burn-in. For the StarBEAST2 analysis, single-gene trees were loaded and visualized by DensiTree [51].

### 2.4. Dating Analysis

Because of the numerous incongruences between the species tree obtained by the multi-locus analyses and the single-gene trees (see Section 3), which could bias the dating analysis, we produced a conservative dataset by: (i) limiting the species sample to 10 representative species (*C. capitata*, *Z. cucurbitae*, *B. bryoniae*, *B. dorsalis*, *B. jarvisi*, *B*. *latifrons*, *B. minax*, *B. musae*, *B. oleae*, *B. tryoni*) and (ii) considering only those 37 genes that produced a ML tree supporting the consensus species tree with minimum ML bootstrap values of 50 at each node.

Divergence times were then estimated by BEAST v. 2.5.1 using the 4-fold degenerate sites of the resulting concatenated dataset (11,768 nt). We calibrated the crown of the Dacinae (*Ceratitis*-*Bactrocera* split) by setting a root prior uniformly distributed between 6 and 65 million years ago, which correspond to the age of a *Ceratitis* fly fossil [52] and of the Schizophora radiation [53,54], respectively. The exact placement of fossils is often disputed and, in general, fossils are also misused because phylogenetic models generally use them to calibrate nodes even though they represent extinct lineages on the stem and not the exact ancestors of the extant ones. In light of this, the mutation rate represents an alternative valid calibration. Indeed, the 4-fold degenerate site dataset also allowed us to use a spontaneous (neutral) mutation rate as an additional prior. Since mutation rate in Tephritidae is not known yet, we assumed it to be similar to that of *Drosophila* (another Diptera) and used the estimate of 0.0346 (SD = 0.0028) substitutions per base pair per million years, provided in [55,56]. This assumption is reasonable, as the mutation rate of insects is quite conserved, with the spontaneous mutation rate of *Heliconius melpomene* (Lepidoptera) being very close to that of *D. melanogaster* (2.9 × 10^−9^ and 3 × 10^−9^, respectively [57,58]). In addition, the large SD derives from the error associated with the mutation rate estimate and, thus, also accounts for the between-branch variation in substitution rates. Because in *Bactrocera* we assumed eight generations per year (in nature they range from 3 to 5 of *B. oleae* and sub-tropical *B. dorsalis* populations, to >12 for the tropical species; [59,60,61,62]) and to account for uncertainty, we finally set as prior a normally distributed mean of 0.028 (SD = 0.03). In a second approach, we set a mutation rate log-normally distributed with “mean in real space” M = 0.028 and S = 0.82 (to produce the same 95% quantile, 0.077, as the normal distribution). For both approaches we performed a model selection to choose the most fitting clock and demographic prior based on the marginal likelihood values with the nested sampling approach implemented in the NS package [63]. We tested the strict and the LOGN relaxed clock and the Yule and the birth–death models, for a total of eight different combinations. Following the recommendations provided by the dedicated Taming the Beast tutorial [50], the sub-chain length was set at 50,000, which corresponds to the length of the MCMC run (i.e., 5 × 10^7^) divided by the smallest ESS value observed across the eight model runs (i.e., ~1000), and the number of particles was set at 10. A model was considered favored over another if the difference between the two marginal likelihoods (i.e., the Bayes factor (BF) in log space) was more than twice the sum of the corresponding standard deviations. We ran eight different analyses that used different combinations of priors and model settings and performed a model selection to identify the most appropriate for our data. In particular, the nested sampling approach allowed us to estimate the marginal likelihoods of the different models and make pairwise comparisons using the associated Bayes factor. All models had marginal likelihoods with a standard deviation ranging from 2.5 to 2.9, which was small enough to assess whether a model was favored over another one. In all cases, we employed a GTR + G replacement model. Because the Bayesian phylogenetic analysis on the concatenated 4-fold degenerate sites resulted in a topology incongruent to the one supported by all other ML and Bayesian analyses (see Section 3), the species tree was fixed according to the consensus topology.

All analyses were performed twice, with chains ran for 5 × 10^7^ generations, sampling trees and parameters every 1000 generations and inspecting convergence and likelihood plateauing in Tracer. Both chains resulted well mixed, with average effective sample size (ESS) values across posterior values being well above 200. The consensus trees (maximum clade credibility trees) were generated after discarding the first 20% of generations as burn-in.

## 3. Results

### 3.1. Phylogenetic Analysis

The results of our analyses strongly indicate that *B. dorsalis* is more closely related to *B. latifrons* than to *B. tryoni* (Figure 1). This relationship is highly supported according to both the ML bootstrap values (≥98, Figure 1a and Appendix A) and the Bayesian posterior probabilities obtained using both BEAST and PhyloBayes analyses (PP = 1, Figure 1a and Appendix A), no matter whether they were based on the codon or aminoacidic alignments. The only support for the relationship ((*B. dorsalis*, *B. tryoni*), *B. latifrons*) comes from the Bayesian analysis run in BEAST2 and was based on the concatenated alignment of the 4-fold degenerate sites (Appendix A). This latter dataset contains sites under (nearly) neutral evolution and, therefore, is suitable for divergence time estimates because it allows a calibration in which a neutral spontaneous mutation rate can be considered equal to the substitution rate [18,56]. This type of dataset is, however, more prone to saturation, and hence, artefacts due to systematic errors. When the 4-fold degenerate site dataset was analyzed in PhyloBayes (which is also a Bayesian implementation) using the among-site heterogeneous CAT model, instead of the among-site GTR homogenous model in BEAST2, we retrieved the same topology obtained by the aforementioned Bayesian and ML analyses on the nucleotide and aminoacidic datasets (Appendix A; see also Appendix A for the results of the ML analysis on the same dataset). Heterogeneous models such as CAT can accommodate for systematic errors related to site-specific variation of the replacement pattern: this suggests that the discordant topology obtained by BEAST2, and also supported by mitochondrial data, is likely an artefact due to model inadequacy.

### 3.2. Dating Analysis

Our dating analysis is based on the best combination of priors according to a model selection whose results (Appendix A) indicated as the favored model the one where we set the mutation prior with a log-normal distribution, a strict clock, and a birth–death model. The Bayes factor values, even after correcting for uncertainty by subtracting the corresponding standard deviations, were well above two, which provides overwhelming support for that model [64]. The fact that a strict clock is favored over a relaxed clock is consistent with the low mean value of the coefficient of variation parameter (i.e., the standard deviation of branch rates divided by the mean rate), which equals 0.24. Therefore, in the main text we report and discuss the results obtained by this analysis.

The clade including *Zeugodacus* and *Bactrocera* split from *Ceratitis* between 6 and 45 million years ago (mean 19 million years ago, mya; 95% highest posterior density, 95%HPD, of 6–45 mya; Figure 2), with *Zeugodacus* then diverging ca. 7.6 mya (95%HPD: 2.3–18 mya). The diversification of *Bactrocera* followed immediately after, ca. 5.9 mya (95%HPD: 1.8–14 mya), with a rapid radiation subtending the species-rich clade [24] that includes the polyphagous species *B. dorsalis*, *B. latifrons* and *B. tryoni* happening between ca. 2.1–1.9 mya (95%HPDs 0.6–5 mya).

## 4. Discussion

### 4.1. Phylogenetic Analyses Reveal a Closer Affinity of B. dorsalis to B. latifrons Than to B. tryoni

The results of the phylogenetic analyses reveal relationships that are in contrast with all phylogenies inferred from mitochondrial sequences (e.g., [30,31]), which support *B*. *dorsalis* as being more closely related to *B. tryoni* than to *B. latifrons*. The results are instead consistent with a closer relationship between *B. dorsalis* and *B. latifrons*, as only partially supported by three previous studies. The first of such phylogenetic analyses, based on 167 Dacini species including *Bactrocera* [24], however, was not conclusive in determining the relationships between these three species, since the phylogeny had many unresolved nodes, including those relative to the most common ancestors of *B. dorsalis*, *B. latifrons* and *B. tryoni*. The low power to disentangle such relationships likely derives from the small dataset—seven nuclear genes—used to produce their phylogeny. A second study by Choo and colleagues [32] analyzed 116 orthologous genes across 11 *Bactrocera* species: despite the larger dataset, their results were also not adequately supported, as the split between (*B*. *dorsalis*, *B. latifrons*) and *B. tryoni* in a ML analysis had a bootstrap value of 70, indicating lack of statistical confidence. We also note that the incongruence does not extend to the species that in our analyses are identified as the closest relatives to *B. dorsalis* (i.e., *B*. *musae*), *B. latifrons* (i.e., *B. bryoniae*) and *B. tryoni* (i.e., *B. jarvisi*), and which are characterized by a longer divergence history (see below). Similarly, a third study [26] that produced an even larger phylogenomic dataset using highly multiplexed amplicon sequencing was not conclusive regarding the relationships between these three species, with a polymorphism-aware phylogenetic model [65] analysis favoring *B. dorsalis* being more closely related to *B. tryoni* than to *B. latifrons*, and an ASTRAL analysis in which this relationship was unresolved.

The discordance between the results obtained by previous studies and our study points to a complex evolutionary history of the group, as suggested by coalescent-aware analyses. Combining single-gene analyses into a coalescent framework also supports *B. dorsalis* as being more closely related to *B. latifrons* than to *B. tryoni*, both when using ASTRAL and StarBEAST2 (Appendix A). It is evident, however, that many genes have phylogenies not consistent with the inferred species phylogeny. For instance, the StarBEAST2 approach reveals a high number of genes having an alternative phylogeny within the ((*B. dorsalis*, *B. latifrons*), *B. tryoni*), suggesting incomplete lineage sorting due to fast radiation or (ongoing) hybridization. For example, in the nucleotide datasets, 58 genes have a ((*B. dorsalis*, *B*. *latifrons*), *B*. *tryoni*) topology, 38 a ((*B. dorsalis*, *B. tryoni*), *B. latifrons*) topology and 14 a ((*B*. *tryoni*, *B. latifrons*), *B. dorsalis*) topology. This uncertainty is also apparent in the ASTRAL results, whereby the support for such a clade falls to <92 when bootstrapping by gene resampling (compare Appendix A). In the light of these results, we cannot exclude the possibility that these species experienced and still experience hybridization events, which could then result in widespread introgression events. Indeed, hybrids have been reported for several closely related *Bactrocera* species [66,67,68,69,70,71,72,73,74,75,76], and although none of the published studies involved the pair of species analyzed in our analyses, possible introgression can occur via direct hybridization or via intermediate hybridization events involving other closely related species. Finally, our results are also consistent with biogeography data: *B. dorsalis* and *B. latifrons* are originally from South Asia, whereas *B. tryoni* is native of Australia, suggesting that their ancestral distributional range also reflects their degree of evolutionary divergence.

### 4.2. Dating Analysis Suggests Fast and Recent Radiation in Bactrocera

Incomplete lineage sorting is expected for rapid radiations (e.g., [77]): this is exactly what is revealed by our molecular clock analyses, which support more recent divergences for the *Bactrocera* radiation compared with previous estimates [23,32,78,79]. Consistent with a rapid radiation of the (*B. dorsalis*, *B. latifrons*, *B. tryoni*) clade, the results of the clock analysis place its origin in the mid-Pliocene, with a mean at ~2.08 mya (95%HPD: 0.6–5 mya), and a subsequent, very close cladogenesis centered at ~1.87 mya (95%HPD: 0.6–4.5 mya) separating *B. dorsalis* and *B. latifrons* (Figure 2; see also Appendix A for the time trees obtained using the different models reported in Appendix A). Interestingly, during this period the sea rose at peak levels [80] and, thus, increased distances between islands and island groups, possibly facilitating allopatric speciation (the three species have native ranges in southeastern Asia and Australia). The proximity of the two cladogenetic events and the large overlap of their 95% confidence intervals agree with a rapid radiation, which could have resulted not only in frequent incomplete lineage sorting but also in possible hybridization events as discussed above. This would also explain the discordant results between the nuclear and the mitochondrial phylogenies, a finding that is reported in many organisms, including insects [81,82,83,84].

Our results also revealed that the split between *Zeugodacus* + *Bactrocera* and *C. capitata* could have occurred between 6 and 45 million years ago (mean of ca. 19 mya). Previously published works proposed more ancient *C. capitata* and *Bactrocera* splits at around 24.9 mya (no confidence intervals reported, [78]), 83 mya (95% highest posterior density: 64–103 mya [79]), and 110.9 mya (95% confidence interval: 91.2–131.4 mya [23]). The difference in the estimates could be due to several factors such as the type of molecular markers (nuclear or mitochondrial), the choice of clock and demographic models, and more importantly, the type of calibrations employed. Here we employed a strategy based on a calibration that combined fossil data and mutation rate, in which we assumed a defined number of generations per year on the basis of available demographic evidence [59,60,61,62]. Generation time can, however, greatly influence the divergence time estimation: for example, if in our analysis we had assumed five generations per year (instead of eight) without changing the other parameters, the mean divergence time would increase from ~20 to ~32 mya for the *Bactrocera*-*Ceratitis* split. The latter is close to what was estimated by Choo and colleagues [32], who performed phylogenetic and dating analyses concatenating nuclear (*n* = 116) and mitochondrial (*n* = 13) genes of a similar species dataset and whose divergence time confidence intervals for this node partially overlap with ours (31.21 mya, 95%HPD: 41.27–21.61; this study: 18.86 mya, 95%HPD: 6–44.98). We also employed a mutation rate prior [55] for the 4-fold degenerate sites that is different and, in our opinion, more conservative than the calibration used by Choo and colleagues. The molecular clock is known to change among sites and lineages [85] and through time [86,87]. Therefore, when choosing a rate prior it is important that it refers to a similar timescale, and especially to sites that evolve similarly. Our approach used a molecular rate estimated from *Drosophila*, which has a generation time and a phylogeny timescale similar to *Bactrocera*; hence, the 4-fold degenerate site rate prior may indeed be a reliable assumption. In contrast, Choo and colleagues used a different approach, whereby they specified as prior only the divergence between *Rhagoletis* (Diptera: Tephritidae) and *Drosophila*, previously inferred in [88].

Finally, we would also like to point out that the mutation rate prior distribution can be a strong determinant of the divergence time estimates and, therefore, need to be carefully tested against alternative models (Appendix A): for example, mean divergence time between *C. capitata* and *Bactrocera* is estimated between 18.3 and 18.9 mya when using a log-normal distribution (Figure 2 and Appendix A), whereas it is estimated between 13 and 13.2 mya using a normal distribution (Appendix A), and the same reduction of ~30% holds for all other divergence times across the phylogeny. Nevertheless, model selection indicates that a log-normal is the most fitting prior distribution of the rate; therefore, the divergence times depicted in Figure 2 should be regarded as more likely. Irrespectively of the divergence estimate for the deep splits, our results clearly indicate that the clade to which *B. dorsalis*, *B. latifrons* and *B. tryoni* belong underwent a rapid and recent diversification.

## 5. Conclusions

Overall, our clock analyses reveal a recent and fast radiation scenario of *Bactrocera* evolution and our phylogeny provides a useful framework for future comparative genomics and comparative biology studies in the major *Bactrocera* pest species. The possibility that hybridization can still occur between closely related species also warns about the possibility that selective events in one species (for instance, resistance to insecticides) may be readily transferred to other species by introgression. More generally, our results once again highlight the importance of comparing different evolutionary models to understand complex phylogenies.

## Figures and Tables

**Figure 1 insects-13-00603-f001:**
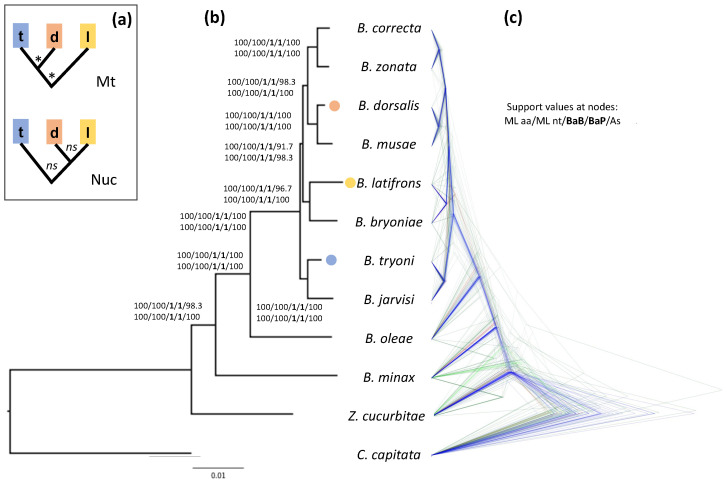
Phylogeny of Bactrocera inferred from the aminoacidic alignments of 110 orthologous nuclear genes. (a) Alternative phylogenetic relationships with supported (*) and non-supported (ns) nodes between B. dorsalis (d), B. latifrons (l) and B. tryoni (t) as presented in the literature and based on mitochondrial (Mt) DNA (e.g., [30,31]) or nuclear (Nuc) DNA (e.g., [24,26,32]). (b) Phylogeny obtained using a maximum likelihood phylogenetic analysis on the concatenated aminoacidic alignments (63,297 aa). Support at nodes is given as bootstrap values for the ML analyses (both for the aminoacidic and nucleotide alignments), bootstrap values estimated by performing 100 multi-locus gene + site resampling using a multi-locus coalescent-aware phylogenetic analysis in ASTRAL across all 110 genes (As) and as posterior probabilities for the Bayesian BEAST2 and PhyloBayes analyses on the aminoacidic dataset (BaB and BaP, respectively). Bactrocera dorsalis, B. latifrons and B. tryoni tips are color-coded as in panel A. (c) Bayesian analysis obtained by StarBEAST2, which employs a multispecies coalescent method to estimate species trees from multiple sequence alignments (i.e., one for each of the 110 orthologous gene sets). For this analysis we used the aminoacidic alignments, linking the site models across the gene sets. Note the numerous discordant gene trees, especially within the B. dorsalis–B. latifrons–B. tryoni clade, compared to the species tree (supported by the gene trees in blue).

**Figure 2 insects-13-00603-f002:**
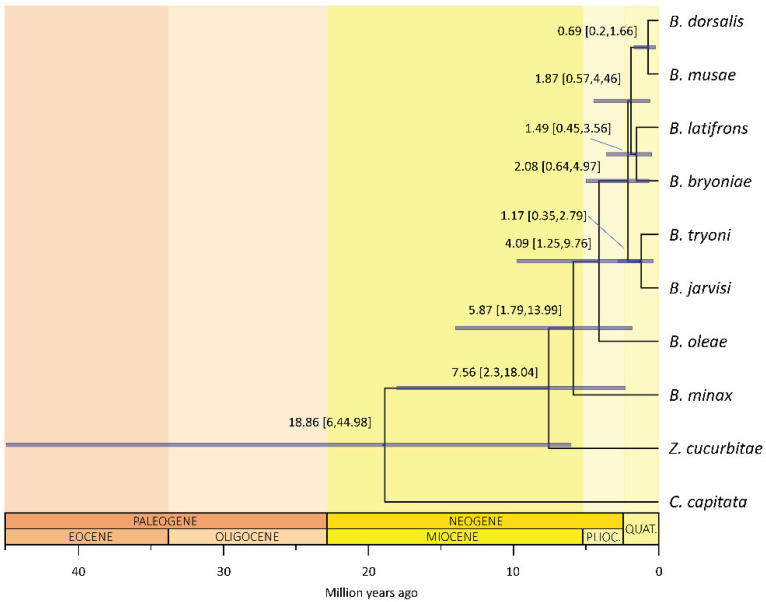
Molecular time tree of *Bactrocera*. *Bactrocera* (plus *Zeugodacus*) likely originated during the Miocene optimum (around 19 mya) and experienced recent, fast cladogenetic events less than 5 mya. The analysis was conducted setting the mutation rate log-normally distributed as prior, a strict clock and a birth–death model. Mean and 95% highest posterior density of the inferred age (blue bars) are reported for each node.

## Data Availability

Publicly available datasets were analyzed in this study. Raw data can be found at https://www.ncbi.nlm.nih.gov/sra accessed on 24 April 2020, with accession numbers indicated in the additional Appendix A, whereas available CDS data can be found at https://www.ncbi.nlm.nih.gov/genome/ accessed on 24 April 2020. The datasets generated and analyzed during the current study, when not available as Appendix A, and the associated scripts are available from the corresponding author on request.

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
