# Peer review of "The Impact of Fast Radiation on the Phylogeny of Bactrocera Fruit Flies as Revealed by Multiple Evolutionary Models and Mutation Rate-Calibrated Clock"

_insects, 2022, doi:10.3390/insects13070603_

Round 1
Reviewer 1 Report
I'm happy with the changes made in this revised version of the ms. No further suggestions from my side.
Author Response
We thank the reviewer for the kind response.
Reviewer 2 Report
The article presented by Valerio and colleagues deals with the phylogenetic relationships within the Bactrocera fruit flies.
Although the introduction is quite short, it makes the point of the paper.
Methods are very well explained. In fact the methodology applied in the paper is excellent in my opinion. It could be said that none novel data is presented, but that is not an issue in this case since the dataset used is re-analysed in a completely new way.
Regarding results and discussion, in my opinion the conclusions are derived from the results. Moreover, the authors are explaining way they get in some genes contradictory results.
For all those reasons I think that the article is very well conceived.
Author Response
We thank the reviewer for the positive comments.
Reviewer 3 Report
Dear Editor,
I read with attention the manuscript titled:
The impact of fast radiation on the phylogeny of Bactrocera fruit flies as revealed by multiple evolutionary models and mutation rate calibrated clock and authored by Federica Valerio, Nicola Zadra, Omar Rota-Stabelli, Lino Ometto
General observations:
The authors have reconstructed the phylogeny of some fruit flies of the genus Bactrocera employing a genome-scaled dataset and different models of molecular evolution. The authors' work focused on the most important species of the genus Bactrocera among which both B. dorsalis and B. latifrons have recently been caught in Europe, in France and Italy the first and only in Italy the second. In the introduction, on the other hand, there is no mention of the recent interceptions which would instead give clear and recent information on the high risk that these species could acclimatize in Europe. This is a flaw that needs to be addressed by the authors.
The authors used sequences made available by other authors and did not do any sequencing work.
The main weakness of the work is precisely this, the use of sequences obtained from others that are not easily identifiable. The sequences used and selected should be put in an additional file to allow the repeatability of the study. It is not clear how the authors selected the sequences and considered the major identification problems that still affect the B. dorsaliscomplex, their results may have been influenced by the use of sequences obtained from bad identifications.
Authors should clarify which sequences are used by clearly identifying them in a supplementary file.
A comparative analysis of the biology of at least the three species (B. dorsalis, B. latifrons, B. tryoni) is missing in the discussions. There is only a brief unclear hint of the different origin.
This part should be improved.
The advancement of knowledge is very minimal.
The manuscript must therefore undergo a major revision to make it suitable for publication.
Minor suggestions
Lines 55-56 Authors should report the recent findings of B. dorsalis and B. latifrons in Europe.
Lines 62-65 and lines 72-74: I believe that the phylogenetic reconstruction of these species is important but these sentences are questionable.
I really cannot understand how the different phylogenetic positions can improve the control of these species. These are mostly polyphagous and multivoltine species (B. dorsalis, B. latifrons, and B. tryoni) whose ranges of hosts overlap. Their control is carried out with identical methodologies, even though the three fruit flies are attracted from different lures B. dorsalis (methyl eugenol), B. latifrons (alpha-Ionol and cade oil,) and B. tryoni (cue lure). Furthermore, the three species have different origins. B. dorsalis and B. latifrons from South Asia while B. tryoni is a native of Australia. Knowing its origin and diffusion, I would expect the allopatric species (B. tryoni) to be more genetically distant from the others.
Line 90: the authorship of B. oleae is Rossi please see Raspi, A., & Viggiani, G. (2008). On the senior authorship of Musca oleae (Diptera: Tephritidae). Zootaxa, 68(1714), 67–68. https://doi.org/10.11646/zootaxa.1714.1.7
Line 92: the Preferred Scientific Name of Zeugodacus cucurbitae is Bactrocera cucurbitae. Please see https://www.cabi.org/isc/datasheet/17683#toidentity.
Line 103 C. capitata should be in italic.
Line 283 B. dorsalis should be in italic.
Lines 334 C. capitata should be in italic.
Lines 336 C. capitata should be in italic.
Lines 347 Ceratitis should be in italic.
Lines 365 C. capitata should be in italic.
The discussions should be improved by making a minimum of biological or ethological comparisons between the main species involved in the analysis.
Author Response
1. The sequences used and selected should be put in an additional file to allow the repeatability of the study. Authors should clarify which sequences are used by clearly identifying them in a supplementary file.
We added supplementary data (Supplementary File S1; see line 115 and 450-454) containing the alignments used to conduct all phylogenetic analyses. See also Table S1 for the SRA accession numbers.
2. Lines 55-56 Authors should report the recent findings of B. dorsalis and B. latifrons in Europe.
We added information regarding Bactrocera dorsalis and Bactrocera latifrons geographical distribution, including recent expansion events. Changes are described in lines 58-61.
3. Lines 62-65 and lines 72-74: I believe that the phylogenetic reconstruction of these species is important but these sentences are questionable. I really cannot understand how the different phylogenetic positions can improve the control of these species. These are mostly polyphagous and multivoltine species (B. dorsalis, B. latifrons, and B. tryoni) whose ranges of hosts overlap. Their control is carried out with identical methodologies, even though the three fruit flies are attracted from different lures B. dorsalis (methyl eugenol), B. latifrons (alpha-Ionol and cade oil,) and B. tryoni (cue lure). Furthermore, the three species have different origins. B. dorsalis and B. latifrons from South Asia while B. tryoni is a native of Australia. Knowing its origin and diffusion, I would expect the allopatric species (B. tryoni) to be more genetically distant from the others.
We agree that the phylogenetic position cannot alone directly help control measures. However, what we stated in lines 62-65 (now lines 65-57) refers to the possibility of learning more about the genetic basis of specific traits that are associated with the “pest” status of a given species, including its host preference (see the very example brought by the referee on the attracting lures, which depend on the chemosensory genes found in the different species, which may be shared or not) or insecticide resistance. This can in turn inform on the possible use of control measures in species for which no field test has been conducted but for which such genetic and phylogenetic information are available. Regarding lines 72-74 (now lines 77-80), we rephrased our sentence to make the statement clearer and less strong. See also issue #12 regarding the last part of the comment.
4. Line 90: the authorship of B. oleae is Rossi please see Raspi, A., & Viggiani, G. (2008). On the senior authorship of Musca oleae (Diptera: Tephritidae). Zootaxa, 68(1714), 67–68. https://doi.org/10.11646/zootaxa.1714.1.7
We revised the authorship from Gmelin to Rossi.
5. Line 92: the Preferred Scientific Name of Zeugodacus cucurbitae is Bactrocera cucurbitae. Please see https://www.cabi.org/isc/datasheet/17683#toidentity.
The taxonomic status of Zeugodacus as a subgenus of Bactrocera was revised in 2015. According to recent phylogenetic studies, the Bactrocera (Zeugodacus) subgenus had to be elevated to genus level because more closely related to the Dacus genus than to the Bactrocera genus.
References
• Krosch MN, Schutze MK, Armstrong KF, Graham GC, Yeates DK, Clarke AR. A molecular phylogeny for the Tribe Dacini (Diptera: Tephritidae): systematic and biogeographic implications. Mol Phylogenet Evol. 2012;64:513–23. doi: 10.1016/j.ympev.2012.05.006.
• Virgilio M, Jordaens K, Verwimp C, White IM, De Meyer M. Higher phylogeny of frugivorous flies (Diptera, Tephritidae, Dacini): Localised partition conflicts and a novel generic classification. Mol Phylogenet Evol. 2015;85:171–9. doi: 10.1016/j.ympev.2015.01.007.
• De Meyer M, Delatte H, Mwatawala M, Quilici S, Vayssières J-F, Virgilio M. A review of the current knowledge on Zeugodacus cucurbitae (Coquillett) (Diptera, Tephritidae) in Africa, with a list of species included in Zeugodacus. In: De Meyer M, Clarke AR, Vera MT, Hendrichs J (Eds) Resolution of Cryptic Species Complexes of Tephritid Pests to Enhance SIT Application and Facilitate International Trade. ZooKeys 2015;540: 539-557. doi: 10.3897/zookeys.540.9672.
• San Jose M, Doorenweerd C, Leblanc L, Barr N, Geib S, Rubinoff D. Incongruence between molecules and morphology: A seven-gene phylogeny of Dacini fruit flies paves the way for reclassification (Diptera: Tephritidae). Mol Phylogenet Evol. 2018;121:139–49. doi: 10.1016/j.ympev.2017.12.001.
• Dupuis JR, Bremer FT, Kauwe A, San Jose M, Leblanc L, Rubinoff D, et al. HiMAP: Robust phylogenomics from highly multiplexed amplicon sequencing. Mol Ecol Resour. 2018;18:1000–19. doi: 10.1111/1755- 0998.12783.
6. Line 103 C. capitata should be in italic.
7. Line 283 B. dorsalis should be in italic.
8. Lines 334 C. capitata should be in italic.
9. Lines 336 C. capitata should be in italic.
10. Lines 347 Ceratitis should be in italic.
11. Lines 365 C. capitata should be in italic.
We italicized all scientific names.
12. The discussions should be improved by making a minimum of biological or ethological comparisons between the main species involved in the analysis.
We think that biological and ethological comparisons go well beyond the scope of this manuscript, which focus on more “technical” aspects of the phylogenetic inference of this group. Other previous works have discussed at length these issues and our findings do not add much more information in this respect. Following a useful comment by the referee (see end of issue #3), we however included (lines 274 and 326-329) some biogeographical and biological information that help interpreting the results of our analyses.
Round 2
Reviewer 3 Report
The authors answered my questions as well as accepted my suggestions of changing some parts or adding some new information on the manuscript. I am satisfied with this second version and I have not objections now to its publication.
This manuscript is a resubmission of an earlier submission. The following is a list of the peer review reports and author responses from that submission.
Round 1
Reviewer 1 Report
The manuscript “The impact of fast radiation on the phylogeny of Bactrocera fruit flies as revealed by multiple evolutionary
models and mutation rate calibrated clock” The authors of this manuscript took previously sequenced data (DNA and RNA) from repositories and pulled out putative 110 single-copy orthologs to construct a “backbone” phylogeny for Dacini. They state that this will “provide a new picture of Bactrocera radiation and can serve as basis for future comparative analyses” However the backbone phylogeny for a group that doesn’t have many species sequenced or specimens that are hard to get can be useful. They fail to explicitly mention in the introduction that these same species were analyzed previously with phylogenomic data with many more samples and other species (878 amplicons from 82 species) in a paper published in 2018 (Dupuis et al. 2018) with very similar results. This the phylogeny they provide is not new to science. One aspect that is new is the attempt to date the phylogeny of Dacini with genomic data. This was attempted before by Krosch et al 2012 with much less data but much more samples and they used outgroup dating. However, the methods used in this manuscript in my opinion are not up to scrutiny. One major issue with the dating that I have is the use of a single calibration point and the mutational derived dating to infer a time tree for Dacini. As the authors state, they used a single uniform calibration point prior on the root “which correspond to the age of a Ceratitis fly fossil (70) and of the Schizophora radiation”. This calibration point severely limits the ages for the root of this phylogeny and doesn’t take into consideration the uncertainty for both of the limits (65mya and 5mya) used to justify this calibration. In my opinion, would be better to use a broad normal distribution prior at 65mya for the root if one would like to date this phylogeny. However, with a single calibration point, I would still say this is not a robust dating method as multiple calibration points would be ideal but most likely impossible with the dataset presented. The other issue I have with the dating in this phylogeny is the use of mutational derived dating. When a highly informative prior has so much influence on the actual dating of the phylogeny, I would not put much confidence in that analysis. Because of these issues, I cannot recommend this paper for publication.
Dupuis JR, Bremer FT, Kauwe A, San Jose M, Leblanc L, Rubinoff D, Geib SM. 2018.
HiMAP: Robust phylogenomics from highly multiplexed amplicon sequencing. Mol Ecol Resour, 18:1000–1019.
Krosch MN, Schutze MK, Armstrong KF, Graham GC, Yeates DK, Clarke AR. 2012. A molecular phylogeny for the Tribe Dacini (Diptera: Tephritidae): Systematic and biogeographic implications. Molecular Phylogenetics and Evolution, 64:513-523.
Reviewer 2 Report
This paper provides a dated phylogeny based on publicly available ‘omic data and aiming at better clarifying the evolutionary relationships between three agricultural pests from the genus Bactrocera (Diptera, Tephritidae).
As it’s often the case, estimates of divergence times based on tentative generation times and spontaneous mutation rates come with considerable uncertainty (as correctly reported by the Authors) and this prevents digging too much into the timeline of radiation of the target taxa. For this reason, I’ve found the second part of the discussion (dating analysis) rather speculative and sometimes potentially misleading (e.g. when stating that “Our results revealed that the split between Zeugodacus+Bactrocera and C. capitata occurred less than 20 million years ago” and not “between 6 and 45 mya” as it would be more correct).
With reference to a previous revision of this ms I did for another journal, I still don't agree about the way divergence time estimates are reported, as I still find that "∼" doesn't provide a fair idea about the extent of the errors associated to the estimates (even when CI are correctly reported somewhere else in the ms text).
In a nutshell: this paper relies on a beautiful introduction and on very solid and clearly presented methods. However I believe that its quality could still improve if the Authors would always and transparently refer to the confidence intervals of their divergence time estimates.
Reviewer 3 Report
It is a robust manuscript from a data analyses and bioinformatics perspective. Unfortunately, from a phylogenetic perspective, the current manuscript cannot answer the proposed questions about Bactrocera phylogeny.
Bactrocera is a genus with more than 450 species, a phylogeny of 10 species of different clades cannot resolve the relationships between species of the genus. In the published phylogeny of San Jose et al. (2018), with a taxonomic sampling of more than 100 Bactrocera species doesn't discuss the species relationships into the genus. The authors affirm:
"Our phylogeny confirms the monophyly of Dacus, Bactrocera, and Zeugodacus. However, most groups below the genus level are not monophyletic, and only through further revision will we be able to understand their evolution and clarify the taxonomy within this tribe."